# A New Fluorescence Detection Method for Tryptophan- and Tyrosine-Derived Allelopathic Compounds in Barley and Lupin

**DOI:** 10.3390/plants12101930

**Published:** 2023-05-09

**Authors:** Sara Leite Dias, Adriana Garibay-Hernández, Fabian Leon Brendel, Benjamin Gabriel Chavez, Elena Brückner, Hans-Peter Mock, Jakob Franke, John Charles D’Auria

**Affiliations:** 1Department of Molecular Genetics, Leibniz Institute for Plant Genetics and Crop Plant Research (IPK), 06466 Gatersleben, Germany; 2Department of Physiology and Cell Biology, Leibniz Institute for Plant Genetics and Crop Plant Research (IPK), 06466 Gatersleben, Germany; 3Institute of Botany, Leibniz University Hannover, 30419 Hannover, Germany

**Keywords:** gramine, hordenine, allelopathy, UPLC, chromatography, fluorescence detection, barley, lupin

## Abstract

Barley (*Hordeum vulgare*) is one of the most widely cultivated crops for feedstock and beer production, whereas lupins (*Lupinus* spp.) are grown as fodder and their seeds are a source of protein. Both species produce the allelopathic alkaloids gramine and hordenine. These plant-specialized metabolites may be of economic interest for crop protection, depending on their tissue distribution. However, in high concentrations they pose a health risk to humans and animals that feed on them. This study was carried out to develop and validate a new method for monitoring these alkaloids and their related metabolites using fluorescence detection. Separation was performed on an HSS T3 column using slightly acidified water-acetonitrile eluents. Calibration plots expressed linearity over the range 0.09–100 pmol/µL for gramine. The accuracy and precision ranged from 97.8 to 123.4%, <7% RSD. The method was successfully applied in a study of the natural range of abundance of gramine, hordenine and their related metabolites, AMI, tryptophan and tyramine, in 22 barley accessions and 10 lupin species. This method provides accurate and highly sensitive chromatographic separation and detection of tryptophan- and tyrosine-derived allelochemicals and is an accessible alternative to LC-MS techniques for routine screening.

## 1. Introduction

Modern cereal agriculture is experiencing novel challenges from multiple biotic and abiotic environmental stresses. As a consequence, crop production is severely affected [1]. To avoid intensive use of chemicals, one solution is to increasingly rely on plants’ specialized metabolites for defense [2]. Plant-derived allelopathic compounds can support crops in the competition against weeds and can confer protection against natural enemies, such as herbivorous insects and pathogens [2,3,4,5,6]. Among these phytochemicals, gramine and hordenine (Figure 1) exhibit promising defensive properties [7,8]. Indole amines occur especially in plants belonging to the Gramineae and Leguminosae [9]. In fact, gramine has been reported in various Poaceae, including *Phalaris arundinacea*, *Arundo donax* as well as in some cereals such as oat and barley [10,11,12,13,14], in addition to some Fabaceae, such as *Desmodium pulchellum* [15] and various lupins [16,17]. Unexpectedly, this alkaloid can be also found in the Silver Maple (Sapindaceae) [18].

The global production of barley (*Hordeum vulgare*) is steadily increasing. Recent reports estimate yields of 160 million tons, cultivated on an area of about 70 million hectares across the world [1]. The interest in this cereal arises from its various uses, ranging from feedstock to cereal grain for human consumption and as the primary malt source for beer production [19]. Among crops, barley demonstrates stronger adaptivity to extreme conditions than other widely used crops and vigor in the competition with other species [5,20]. Along with the benefits conferred by gramine and hordenine, the plant relies on an early development of biomass and the production of waxes to withstand environmental challenges [5].

Lupins are gaining more popularity in recent years because, in addition to other nutritional benefits such as antioxidants and dietary fiber, they have high protein content. Lupin-derived products include mashed bean flour, yogurts and milk substitutes [21,22]. Moreover, they are becoming more prevalent in products, such as cereal mixtures, flakes, cookies and protein bars. Lupins, consisting of more than 500 species, are grown worldwide [17,23]. They exhibit good adaptation to adverse conditions, being capable of efficient nitrogen fixation and survival on phosphate-deficient soils [24]. Moreover, their addition in crop rotation systems increases the yield of the following cultivated crop [25].

The amino acid tryptophan is the primary metabolic substrate for the production of gramine in barley. Studies using radiolabeled precursors proposed 3-aminomethylindole (AMI) and *N*-methyl-3-aminomethylindole (MAMI) as intermediates in the pathway of biosynthesis of this alkaloid (Figure 1) [26,27,28,29]. Hordenine derives from the modification of tyrosine which is converted to tyramine via decarboxylation and finally to hordenine via two rounds of methyl transfer (Figure 1) [30,31,32].

Hordenine and its precursor *N*-methyltyramine are present in greater quantities in the roots, whereas gramine accumulates mainly in the leaf tissue [14,33]. In comparison to hordenine, gramine demonstrates higher toxicity [34]. These alkaloids, when present in forage plants, can reduce palatability in mammals [35], while higher concentrations can cause reduced weight gain, kidney lesions with consequent glycosuria or even death in both humans and animals [36,37,38,39]. Furthermore, alkaloids impair the fitness of herbivorous arthropods, such as grasshoppers and aphids, inhibiting their growth and acting as a feeding deterrent [40,41]. Additional organisms affected by gramine include bacteria, algae and fungi [42,43,44,45]. Gramine and hordenine may be interesting for crop protection; however, high concentrations of these compounds pose a health risk to humans and the animals that forage on them. Given our interest in studying the biochemistry and chemical ecology of allelopathic compounds in agriculturally important crop species, we found it necessary to develop an accessible, accurate and highly sensitive method for the separation and detection of tryptophan- and tyrosine-derived metabolites.

In the past, several methods were applied for the analysis of gramine: from paper chromatography [35] to silica gel thin-layer chromatography (TLC) [46] to more recent techniques, including high-performance liquid chromatography coupled with mass spectrometry (LC-MS) [47,48], gas chromatography mass spectrometry (GC-MS) [11,49] and others; but, no method for the analysis of gramine assisted by fluorescence detection has been described yet. Our main motivational force for developing such an option for the analysis of gramine, hordenine and their related metabolites was to provide an affordable and more accessible alternative to mass spectrometry-based analysis. In fact, many agricultural labs are not equipped with LC-MS analytical instruments, whereas fluorescence detection coupled instruments are more prominent in labs, they are cheaper and do not require high maintenance. Therefore, we developed a reliable and accurate ultra performance liquid chromatography (UPLC) method that presents both a new detection step and a chromatographic separation step for the quantification of the mentioned compounds. Our method displays a balanced retention of polar and hydrophobic molecules, providing a solution to the intricate methodological problem linked to the separation and detection of tryptophan- and tyrosine-derived compounds caused by the co-analysis of zwitterionic precursors, sensitivity to small pH shifts, as well as the challenge of achieving baseline separation of nearly structurally identical molecules. Moreover, the sensitivity of the fluorescence-based detection results are comparable to that of certain types of LC-MS while being a cheaper option. We validated the UPLC-FLD method, also providing a comparison between the UPLC-FLD and UPLC-MS performance. Finally, we quantified gramine, AMI, tryptophan, hordenine and tyramine in various tissues of lupins and several barley elite varieties, and reported them with a particular focus on gramine.

## 2. Results

### 2.1. Method Development

Because of our interest in studying the allelopathic compounds of barley plants, we developed a method to separate and detect gramine and hordenine that could allow us to observe the accumulation of these compounds, along with that of their precursors, in various plant tissues. Some other metabolites, such as noradrenaline and dopamine, were added to the standard mix because they are related to the presence of some alkaloids in temperate cereals and their response to biotic stresses. Many stationary phases were tested to achieve the satisfactory separation of compounds (BEH-phenyl, CSH phenyl-hexyl, C18, Restek Raptor fluoro-phenyl and more), along with several mobile phases (methanol, ammonium acetate for alkaline pH of 10, etc.) tested with various elution gradients. A column HSS T3 (100 A, 1.8 μM, 2.1 × 100 mm) was initially selected for the tests, and the standards (0.1 mg/mL) were dissolved in 80% methanol. The mobile phases were water with the addition of 0.1% of formic acid (pH 2.75) (A) and acetonitrile with the addition of 0.1% of formic acid (B), flowing at a 0.45 mL/min rate. Despite showing satisfactory retention of non-hydroxylated tryptophan derivatives, the peak shape requirements were not met along with the coelution of most tyrosine derivatives (Appendix A). Moreover, the alkaline pH of the mobile phase was causing drifting, confirming the column’s instability at high pH. The increase in the formic acid ratio to 0.5% in both the eluents did not improve the separation and elution of the alkaloids with the HSS T3 column (Appendix A). A further attempt was performed with the HSS T3 column, accompanied by the mobile phases: 10 mM ammonium acetate (pH 5.0) (A) and acetonitrile (B). Despite the improved separation of the tyramine derivatives (TYR, MeTYR, HOR) from the DOPA derivatives, the higher retention times and a further improvement in the peak shape of some compounds, non-desirable two-peak elution was still observed (Appendix A).

The tests performed with CSH phenyl-hexyl (130 A, 1.7 μM, 2.1 × 100 mm) coupled with mobile phases consisting of water with the addition of 0.5% of formic acid (pH 2.4) (A) and acetonitrile with the addition of 0.5% of formic acid (B) and also water with the addition of 0.5% of formic acid (pH 2.4) (A) and methanol with the addition of 0.5% of formic acid failed because the compounds were not separating properly, while some were eluting in two distinct peaks (Appendix A). Increasing the proportion of the aqueous mobile phase in the isocratic flow and lowering the flow rate slightly increased the retention and separation between the tested alkaloids, but the peak shape of several compounds appeared to be distorted (Appendix A). An Acquity UPLC BEH phenyl 130 A, 1.7 μM, 2.1 × 100 mm was also tested in the attempt to increase the retention and separation of these alkaloids. Compared to the phenyl-modified CSH phenyl-hexyl phase, this column showed a higher retention; however, the peak shape and compound elution were again highly distorted and the tyramine derivatives eluted as two peaks (Appendix A).

After the assessment of the several aforementioned stationary phases (Appendix A), we selected the HSS-T3 for further optimization of the method by fine-tuning the pH of the mobile phase and the elution gradient (Appendix A). Steeper gradients after the elution of tryptophan (TRY) improved the peak shape and separation, while a slightly acidified water/acetonitrile showed the best separation of the target compounds, and the addition of formic acid ameliorated the peak shape.

The idea to use fluorescence detection of gramine was conceived later. Given that certain proteins and some indole-derived compounds are detectable via photodiode array detector (PDA) at 280 nm, we tried to detect gramine, hordenine and their related metabolites with an excitation wavelength of 280 nm and an emission of 320 nm. Gramine exhibited a strong fluorescence signal corresponding to emission at 319.2 nm, whereas that of hordenine was weaker and variable (Appendix A). The length of the run, comprising the washing and reequilibration phases were studied to achieve sufficient cleaning of the column, avoiding carryover of the analytes.

### 2.2. Selectivity, Accuracy, Matrix Effect and Precision

An unidentified substance in the formic acid was exhibiting some fluorescent activity, leaving minor traces at retention time 6.39 ± 0.37 min, which merged with the peak of gramine when present on both instruments (1) and (2). These traces contributed an average peak area of 234,431 to the background (1) (Appendix A) which, in an equivalent gramine amount, accounts for (0.460 pmol/µL/mg FW). No other endogenous substance was found interfering with the analysis of the analytes contained in both the standard mix and tested plant tissues. The peaks corresponding to the abovementioned molecules were well separated and matched different retention times (Figure 2, Appendix A).

The accuracy was expressed as the difference between the theoretical injected concentrations and those measured with confidence intervals of ± 1 standard deviation (SD). Standard injections showed 99.9 to 108.3% of the intended injections of 8, 25 and 100 pmol/µL. Repeated spiking of barley kernel extracts, var. Golden Promise, showed a maximum relative standard deviation (RSD) of 2.3% and the results diverged from the value of the standard injection to the extent of 97.8–112.8%; however, the spiking of the var. ZDM 01467 kernel extracts showed a variability corresponding to a maximum 2.0% RSD and compared to the standard it contained 15.3 to 23.4% more gramine (Table 1). The control samples contained slightly increased concentrations of gramine in comparison to the standard injections.

The precision of the method was evaluated by determining the RSD for the technical replicates of the lupin pooled seed samples. The samples varied between 93.3 and 105.7% compared to the average value. The RSD was 0.8, 2.8 and 6.3 for the samples of *L. luteus*, *L. angustifolius* and *L. mexicanus*, respectively (Table 2).

### 2.3. Calibration Curves and Sensitivity

Standard curves were plotted as a concentration (µM) vs. response (peak area) plot. The calibration curves of the gramine, hordenine, AMI, tryptophan and tyramine obtained from the LC-FLD (1) met the linearity requirements as they fitted well to the equation by linear regression in the concentration range 0.09–100 pmol/µL. The injection < 0.2 pmol/µL did not produce a signal-to-noise ratio that allowed for the visual identification of a corresponding peak. The “y = mx + b” equation and the coefficient of determination for the five analytes are reported in Table 3. The equations obtained via LC-MS (1) are reported in Appendix A. Instrument (2) showed a wider range of linearity, namely 0.50–900 pmol/µL, but had slightly higher limits of detection and quantification. The equations of the calibration curves generated on (2) are reported in Appendix A. The samples whose metabolite’s concentration were exceeding those of the linear range had to be further diluted to be analyzed.

*Response standard deviation and slope* was the method used for the estimation of the limit of detection (LOD) and limit of quantification (LOQ) [50]. The LOD and LOQ for gramine, hordenine, AMI, tryptophan and tyramine on UPLC-FLD and LC-MS are reported in Table 4. The calculated LOD of gramine was 0.09 pmol/µL, which nicely matched the visual evaluation as the first peak discernible from the baseline noise was the one generated by the injection of 1 pmol/µL. Injections ≥ 500 pmol/µL caused column saturation resulting in blunt peaks. The LOD of gramine was equal to 0.26 pmol/µL. For the same calibration runs, we measured an LOD of 0.27 pmol/µL and an LOQ of 0.83 pmol/µL for gramine on the LC-MS. The LOD and LOQ values of the five analytes measured on instrument (2) are listed in Appendix A and are reported in Figure 3, Figure 4 and Appendix A as a dashed green line and a red line, respectively. 

### 2.4. Carryover

In regard to carryover, no analyte remnants deriving from plant tissues nor standards were identified in the preceding chromatograms. This was tested by inserting blank ACN runs after injections performed to determine the upper limits of quantification (ULOQ). An unidentified contaminant in the formic acid generated a response upon fluorescence detection, contributing to a high signal-to-noise ratio. Discernible peaks in the washes corresponded to traces left by this latter compound (Appendix A).

### 2.5. Stability

As for stability and reproducibility, we did not detect important variation in the results depending on the instrument or its components, nor in the stability of the extracts after storage (conditions described in the methods) for longer periods of time. Variability in the results was within the values reported in Section 2.3 upon the method’s precision assessment. Regardless, we recommend initiating the instrument long before the start of the analysis to give it time for needed equilibration. 

### 2.6. Applications

The method was used to conduct a study on the gramine content in various accessions of barley, (*Hordeum vulgare*), as well as in several species of lupins (*Lupinus* spp.). The analysis also detected further allelopathic compounds and precursors, namely hordenine, tryptophan and tyramine. The inspected tissues included kernels, leaves and root material for barley and seeds and leaves tissues for lupins.

#### 2.6.1. Screening the Barley Pangenome

Gramine production was highly variable among the 22 screened accessions of barley. Lines Golden promise, Igri and Morex had neglectable amounts of gramine as well as the value of ZDM 01467 which was close to the LOQ. The gramine concentration found in the leaves (8 DAG) of the remaining lines varied from 1184.17 to 6579.68 pmol/µL on average (Figure 3a). No gramine was quantified in the kernel samples except for the seeds of HOR 3365 that contained 66.42 pmol/µL (Figure 3b). In the roots (8 DAG), gramine was below the limit of detection (Appendix A), whereas hordenine was expressed in the range 27.60–580.60 pmol/µL among all the barley accessions, except for lines B1K-04-12, HOR 21599 and HOR 7552, where the amounts did not reach the limit of quantification (Figure 3c). Not all B1K-04–12 seeds germinated; for this reason, we had two measurements for each timepoint for the assessment of the metabolites in both the leaf and root tissues. The measurements of all the investigated metabolites in the leaf, root and kernels are reported in Appendix A.

#### 2.6.2. Gramine in Lupins

Gramine was found in the pooled seeds samples of *L. mexicanus*, *L. luteus* and *L. angustifolius* which reported the highest gramine concentration, namely 3009 pmol/µL on average (Figure 4a). The *L. hispanicus* seeds were not analyzed due to seed unavailability, whereas the analysis performed on the leaves of the lupin species revealed average gramine levels of 2309 pmol/µL. High concentrations of gramine were also found in the leaves of *L. mexicanus*, and *L. luteus*, 3605 and 515.4 pmol/µL, respectively, but in contrast to the seeds, the leaves of *L.angustifolius* only contained traces of it (Figure 4b). The leaves of *L. atlanticus* could not be collected as no seeds germinated, while the data reported for the metabolites present in *L. consentinii* and *L. mutabilis* only refer to one event of leaf collection as only one plant per species was available at the time. The measurements of all the investigated metabolites in the seeds and leaves are reported in Appendix A.

## 3. Discussion

In this study, we report a new chromatographic method for the separation of gramine and hordenine that proposes fluorescence as a valid alternative technique of detection for these compounds and their related metabolites. The developed method was consecutively validated and the reported values for accuracy, precision, linearity and those of the LOD and LOQ were evaluated as acceptable.

Accurate chromatographic separation was achieved after testing various combinations of different stationary phases and eluents. The UPLC HSS T3 column resulted in being the suitable choice for this application as it is a low ligand density C18 column that shows enhanced retention of polar compounds and metabolites by reversed phase liquid chromatography, and displays a balanced retention of polar and hydrophobic molecules [51,52]. Moreover, it contributed to a time of analysis of 10.5 min, which is relatively short compared to previously described runs lasting more than 30 min, and it is comparable to the shorter times of 6 min runs performed by Khedr et al. on LC-MS/MS [53,54,55]. The precision of the method for the measurement of gramine was assessed on pooled seeds of lupin. It accounted for 93.3–105.7% of the average amount, whereas the relative standard deviation (RSD) was <7%, namely 0.8, 2.8 and 6.3 for the samples of *L. luteus*, *L. angustifolius* and *L. mexicanus*, respectively. The increased concentration of gramine in the control samples (>100%) might be dependent on the matrix effect: plant metabolites might interact with gramine’s detection during chromatographic analysis. The accuracy was calculated using control samples which consisted of kernel extracts of two varieties of barley—which did not contain detectable levels of gramine during the screening. These samples were subsequently spiked with known concentrations of gramine (8, 25 and 100 pmol/µL). The accuracy was 97.8–112.8% for var. Golden promise and 115.3–123.4% for var. ZDM 01467. Pharmacokinetic studies on hordenine concentrations in rat plasma with UPLC-MS/MS reached accuracy and precision values of 80.4–87.3% and < 8% RSD, respectively [56]. Compared to these latter values, we obtained an excess of gramine rather than a loss during the analysis. We hypothesize that the matrix effect can influence the detection of plant allelopathic compounds: metabolites might interact and impair gramine’s detection during analysis. Considering that both accuracy and precision are subjected to further variables such as the day of analysis, the operator and the instruments, we conclude that our method reaches excellent accuracy and precision in the separation and quantification of gramine levels in seeds and vegetative tissues [57]. 

The established LOD and LOQ of 0.09 and 0.26 pmol/µL fit in the range of linearity. The UPLC-FLD (1) supported the analysis of biological samples whose concentration fell within the range 0.09 to 100 pmol/µL. Injections exceeding 500 pmol/µL of gramine caused column saturation which led to blunt peaks. Additional injections in the range 100–500 pmol/µL would have determined if the range of linearity had a higher upper limit. On our standalone UPLC-FLD instrument (2), injections of standards whose concentrations ranged from 0.5 to 900 pmol/µL resulted in fully symmetric peaks. This range of linearity resulted in being nine times larger than that of the analyses performed on the first instrument equipped with a fluorescence detector (1), despite an increase in the LOD for gramine.

Measuring the limits of detection of different instruments allowed us to compare the sensitivity of fluorescence-mediated detection to more traditional techniques used to measure gramine. Our results showed that it can be as sensitive if not even more accurate than LC-MS. The LOD assessed on the Acquity UPLC system coupled with the fluorescence detector resulted in an LOD corresponding to the abovementioned concentration of 0.09 pmol/µL, whereas with UPLC coupled with an electrospray ionization-ultra-high-resolution quadrupole time-of-flight mass spectrometer, the LOD was determined to be in the range of 0.27 and 0.18 pmol/µL for the fragment ions 130.0669 and 175.1239, respectively (Appendix A). Consequently, our results showed that the detection of gramine mediated by fluorescence was up to three times more sensitive than the one measured via mass spectrometry on the same instrument. Indeed, fluorescence detectors can report higher sensitivity and be preferred over MS- or UV-assisted analysis for the quantification of plant metabolites, environmental toxins and pharmaceutical molecules [58,59,60,61].

In the past, the advent of TLC shortened the development times compared to paper chromatography and reduced the diffusion of the chromatography compounds. TLC improved the sensitivity of detection of synthetic and natural indole-derived compounds by 20-fold. Reagents such as the Renz and Loew reagent, Adamkiewicz reagent (formaldehyde-HCl) and Maickel and Miller reagent (o-phthalaldehyde-HCl), which cause indole condensation products to fluoresce, have high sensitivity of their detection. Unfortunately, the downsides of TLC methods included the fact that they can fail to precisely identify indole derivatives in plant extracts as other non-indolic fluorescing substances also exhibit yellow-orange fluorescence [46]. For a long time, TLC techniques coupled with Ehrlich and van Urk reagents were the most specific for the development of indole-derived compounds, but it still lasted up to 8 h and the resulting colors were unstable because of the degradation caused by the acidic conditions. The lower limit of detection for gramine with this method, estimated by Ehmann in his study, was in the range of 50 ng [46]. Our technique is in the range of three orders of magnitude more sensitive for the detection of gramine compared to the values reported in the literature for thin-layer chromatography, where the concentration and intensity of the color trace left on the plate were found to be in a linear relationship in the range between 0.2 and 2 nmol. Moreover, the current method avoids the problem of other closely related compounds showing similar coloring on the TLC plate, not to mention that it does not require significantly toxic reagents [62].

Liquid chromatography techniques have ameliorated the recovery of alkaloids during analysis. In 1992, Muir et al. published a quantitative high-performance liquid chromatography method for the analysis of gramine in cereals. The method was based on a simplified methanol/ammonia extraction procedure followed by the analysis by reversed phase liquid chromatography coupled with a photodiode array detector (PDA) that enabled scientists to discern gramine from AMI and MAMI. The recovery of gramine was estimated in the range of 76–78%, corresponding to 17–871 mg/mL [53]. The most recent studies suggest water/acetonitrile extraction followed by an LC-MS/MS analysis. This method reports linearity in the range of 0.55–55 mg/kg, an estimated accuracy of 93% and a relative standard deviation ((RSD) < 3.6%) [55]. Concerning the methods’ sensitivities, Ganzera et al. presented LOD values ranging from 24.575 to 129.93 pmol/μL [63] for gramine and other alkaloids present in lupin (such as 13α-Hydroxylupanine and sparteine). The LOD measurement that we collected reports lower values, more comparable to those collected with the more recently developed gas chromatography with flame ionization detection method reported on, for which the LODs ranged in the interval 2.99–4.50 pmol/μL for the same quinolizidine alkaloids observed by Ganzera et al. [64]. Recent updates in the LC-MS/MS analysis of gramine present detected concentrations in lupins as small as 4.5 μg Kg^−1^, i.e., 0.030 pmol/μL [65].

Our analysis of gramine in various accessions of barley, as well as in several species of lupins highlighted how diverse the range of concentration can be within the same species and across plants belonging to the same family. Among the 22 accessions of barley analyzed, the only kernels containing gramine were those of HOR3365. The sample set, which consisted of a pool of seeds/kernels, contained an average of 66.42 pmol/µL/mg FW of gramine. High transformation efficiency lines Golden Promise and Igri, as well as Morex and ZDM0167 contained low levels of gramine in their leaves, whereas varieties Barke, Franka, Hockett and Scarlett did not report any [66,67,68]. Three barley lines contained gramine concentrations exceeding 5000 pmol/µL/mg FW, namely HOR 21599, HOR 3365 and OUN333. No hordenine was detected in the leaves, just as the presence of gramine was not detected in the roots of barley plants, consistent with previously reported results [69]. Hordenine accumulation across the analyzed lines showed variability: in the root material collected, 8 DAG, lines B1K-04-12, Barke, Franka, Hockett, RGT planet and Scarlett did not contain detectable hordenine levels, whereas lines HOR 8148 and Morex exceeded 1000 pmol/µL/mg FW. Previous studies report gramine concentrations exceeding 2000 mg/g dry weight in seedlings of barley and <30 and up to >10,000 μg/dry weight in the shoot of several analyzed barley genotypes belonging to both the H. vulgare and Hordeum spontaneum species [70]. These concentrations are significantly higher than what we detected in this current study. However, the observation that total concentrations of allelopathic compounds decrease with the age of the plants is consistent between our study and previous reports.

Among the 12 species of lupin whose seeds we analyzed, only 4 had detectable amounts of gramine, ranging from the lower amounts found in *L. succulentus* (34.65 pmol/µL/mg FW on average) to the maximum concentration reached by extracts of *L. angustifolius* (3010 pmol/µL/mg FW on average). Intermediate values were detected in the yellow lupin (*L. luteus*, 648.5 pmol/µL/mg FW) and in *L. mexicanus* (241.6 pmol/µL). No hordenine nor tyramine were detected in the seeds, but 247.4 pmol/µL/mg FW of hordenine and 654.0 pmol/µL of tyramine were detected in the leaves of *L. luteus* and *L. mexicanus*, respectively. The gramine assessment in lupin vegetative tissues revealed that some carry traces, namely *L. cosentinii* with 14.09 pmol/µL and *L. mutabilis* with 13.90 pmol/µL/mg FW, whereas *L. hispanicus* and *L.mexicanus* produce as much as 2309 and 3605 pmol/µL/mg FW. The accumulation of secondary metabolites in lupins can vary widely depending on the species, cultivar and growing conditions. Species such as the white lupin (*Lupinus albus*), blue lupin (*L. angustifolius*), yellow lupin (*L. luteus*) and pearl lupin (*L. mutabilis*) have been more extensively studied as they are of higher agroeconomic interest [71]. An analysis of various accessions of *L. luteus* has underlined how, within the same species, a wide range of variation in gramine concentrations can be detected in seeds and leaves [72]. Moreover, further traits can contribute to the concentration of alkaloids in plant tissues, for example, the quality and storage of the lupin seed [65].

The differential production of gramine across tissue types can be explained via spatial differences in the distribution of its precursors, which also applies to hordenine and its precursor tyramine. Because of the localization of their precursors, a greater quantity of gramine is accumulated in the leaves and of hordenine in the roots [54,73]. In barley, the enzyme that in barley leads from AMI to the synthesis of gramine was characterized by Larsson et al. [20]. This *N*-methyltransferase is responsible for the double methylation that converts AMI into MAMI and the latter one into gramine, whereas the gene encoding the conversion of tryptophan to 3-aminomethylindole (AMI) remains to be elucidated. The double methylation converting tyramine into hordenine has not yet been described and it is still unknown if the two steps are performed by the same enzyme [32].

The high variability in gramine’s range of concentration among different barley accessions is indeed not surprising. Barley is one of the most genetically diverse cereal grains [19]. RNA sequencing and genomic similarity analysis have tackled the history of the domestication of this crop and determined its polyphyletic origin [74]. Moreover, gramine is found in several other species belonging to the Poaceae, such as *P. arundinacea*, *P. tuberosa* and *A. donax* [11,35,75], but according to Kokubo et al., gramine biosynthesis is limited to a number of groups. In fact, the Poaceae is large and contains more than 12,000 species divided into 12 subfamilies. The Poeae, the Triticeae and the Arundinoidae tribes generally produce gramine [76,77]. Interestingly, the first two taxa are part of the larger Pooideae subfamily, but the Arundinoideae are part of the PACMAD clade (Panicoideae-Aristidoideae-Chloridoideae-Micrairoideae-Arundinoideae-Danthonioideae) which split from the BOP clade (Bambusoideae, Oryzoideae, Pooideae) 53–61 million years ago [78]. In light of this, and dating the monocot-dicot divergence back to 140–150 Myr ago [79], it is probable that the production of gramine in both barley and lupin plants represents a case of convergent evolution, in which the two taxa have evolved independent pathways for the biosynthesis of the same defensive compound.

In nature, gramine shapes plants’ interactions with the environment, defending them against herbivore insects such as aphids [70,80] and modulating bacterial communities surrounding the plants [81]. A gramine concentration of 0.1% causes reduced survival in cowpea aphids and 100% death by green peach aphids [82], while levels of 40 mg/kg orally administered causes toxicity in sheep [37]. Gramine and other alkaloids (such as piperidine derivatives) present in lupin and barley also raise major concerns for human health and nutrition [80]. High alkaloid concentrations (200 mg/kg for animals and 100 mg/kg for humans) can cause acute anticholinergic toxicity and poisoning [55]. In fact, it is worth noting that gramine, in lupin plants, is equal in concentration to the total amount of quinolizidine alkaloids such as lupanine, sparteine, 13-hydroxylupanine and multiflorine found in the plants [17]. These lysine-derived compounds, accumulating in the range of 5–200 mmol/kg in various plant tissues, protect them by exhibiting even greater toxicity than indole alkaloids [83,84]. Therefore, much effort is focused on elucidating the pathways of the biosynthesis of quinolizidine alkaloids [72]. Meanwhile, lupin-based food products remain subject to regulations and safety assessments in many countries.

Gramine analysis is imperative because as much as gramine is an attractive compound with valuable antifeedant properties, its presence becomes disadvantageous when in plant tissues destined for human and animal consumption, as its concentration is so high that they become toxic. In our study, we report that fluorescence detection resulted in being more sensitive than orthogonal TOF mass spectrometry while maintaining a relatively large linear dynamic range. The liquid chromatography method with fluorescence-mediated detection that we provide offers a wide range of applications. It allows for the detection of gramine in routine food testing for the detection of toxic compounds and can support further functional studies and future agricultural applications related to gramine production such as its manipulation for the production of better forage for livestock. Future studies of this allelopathic compound, comprising the understanding of the metabolic pathways that lead to its biosynthesis in various plants and its role in the plant-herbivore/pathogen interaction, will be crucial, especially in the context of climate change.

## 4. Materials and Methods

### 4.1. Plant Material

The kernels of the different varieties of barley were a gift from Prof. Dr. Nils Stein (IPK, Gatersleben, Germany). To ensure good representation of the genetic diversity of the crop, the following lines were selected for our study: Akashiniriki, B1K-04-12 (a wild variety collected in Israel), Barke, Franka, Golden Promise (oftentimes used in transformation protocols), HOR 10350, HOR 13821, HOR 13942, HOR HOR 21599, HOR 3081, HOR 3365, HOR 7552, HOR 8148, HOR 9043, Igri, Morex (the reference cultivar), OUN333, Hockett and RGT Planet (two varieties used in malting), Scarlett, ZDM01467 and ZDM02064 (two barley lines used by Chinese breeders). More information on the pedigree of the latter one can be found in Jayakodi et al. [85]. Lupin varieties were chosen based on the availability of the IPK genebank repository (IPK Genebank, Gatersleben, Germany) (Table 1).

Lupin seeds (Table 5) were provided by the IPK Genebank (Gatersleben, Germany). 

### 4.2. Plant Growth Conditions

Barley plants were grown in a soil mixture, composed of IPK compost, substrate 2 and white TORF, in Ø 9 cm pots and located in a climatized chamber at 45% humidity, 20 °C and 14 h light/10 h dark photoperiod, light intensity 240 µmol m^−2^ s^−1^. The first batch of barley plants was collected 8 days after germination (DAG), once the second leaf was fully developed. A second collection was performed on another batch 3 days later (11 DAG). On both occasions, the roots were also collected.

Lupin seeds were disposed in a Petri dish with a filter paper soaked with water until germination and then transferred to Ø 11 cm pots filled with the abovementioned soil. Lupin plants were grown under the same climatized room conditions as described above. Collection of the first and second leaves of the plants was performed as soon as the plant had fully developed the fourth leaf.

### 4.3. Sample Preparation

#### 4.3.1. Barley Sample Preparation

The extraction protocols were modified based upon the protocol of Muir et al. [53]. The modifications were the following: Leaf and root tissues were harvested into 2 mL Eppendorf tubes and flash-frozen in liquid nitrogen. Subsequently, 3 steel beads (Ø 3 mm) were added to each tube and the plant material was ground for 45 s at 30 Hz with a tissue homogenizer (Retch MM400, Retch GmbH, Haan, Germany). Additional liquid nitrogen was added to the samples and the grinding repeated. The beads were removed and ~50 mg of the frozen samples were weighed. All solvents used were obtained from Carl Roth GmbH + Co. KG, Karlsruhe, Germany.

Barley kernels were placed in a steel container with a single steel bead (Ø 1.8 cm) and ground for 2 min 30′. A total of 50 mg of material was then collected in a 2 mL Eppendorf tube (pooled samples). Tubes were kept in liquid nitrogen during this process. All samples were stored at −80 °C until extraction. Extraction phase was initiated with the addition of 5 µL pure LCMS grade methanol/mg fresh weight sample. The Eppendorf tubes were vortexed and placed at 4 °C overnight. The following day, samples were centrifuged at 15,000 rpm and 4 °C for 10 min. The supernatant was transferred to new tubes. Leaf and root tissues were resuspended in methanol, whereas kernel extracts were resuspended in 0.5 Vol. CHCl_3_ and 1 Vol. LCMS-grade water and the samples were left at 4 °C for 2–3 h. The centrifugation was repeated as well as the supernatant transfer, which was added to the one isolated previously. Extracts were stored at −20 °C until analysis. Before injection, all samples (leaf tissue, root tissue and kernels) were centrifuged at 15,000 rpm and 4 °C for 10 min and diluted (1:10) with a 0.1% formic acid solution and left to stand for 2–3 h before centrifugation. The supernatant was finally transferred into UPLC vials.

#### 4.3.2. Lupin Sample Preparation

The preparation of lupin samples was similar to the one of barley. For the collection of both seeds and leaves, samples were placed in a steel container with a single steel bead (Ø 1.8 cm) and ground for 2 min 30′. A total of 50 mg of material was then collected in a 2 mL Eppendorf tube. Seed samples were made out of a pool of seeds. An aliquot of 250 µL of methanol was added for every 50 mg of leaf sample, whereas for seeds we added 200 µL for every 50 mg weighed. Seed samples were then sonicated for 10 min at room temperature. Samples were then left at 4 °C until the next day. Half the volume of chloroform was added to the seed sample together with 1 volume of water (LCMS grade), whereas leaf tissue was resuspended in methanol. Samples were vortexed for 1 min and centrifuged at 15,000 rpm for 10 min. A total of 450 µL of the upper phase was transferred into new tubes. Extracts were stored at −20 °C until the day of analysis. Before injection, leaf samples were diluted 1:10 with 0.1% formic acid (20 µL sample and 180 µL of 0.1% formic acid) left to stand for 2–3 h. All samples were centrifuged and the supernatant transferred into UPLC vials.

### 4.4. Chemicals and Reagents

Gramine, hordenine, tryptophan and dopamine hydrochloride were purchased from Sigma-Aldrich (Steinheim, Germany). 3-aminomethylindole was purchased from Fluorochem (Hadfield, UK), DL-Noradrenaline was purchased from Fisher Scientific (Schwerte, Germany), whereas *N*-Methyltryptamine was purchased from Carbosynth (Compton, UK). Acetonitrile, methanol and ultra-pure water were purchased from Chemsolute (Renningen, Germany), formic acid from J. T. Baker (Gross Gerau, Germany) and chloroform from Roth (Carl Roth GmbH + Co., Karlsruhe, Germany). 

### 4.5. Instrumentation and Conditions

The samples were analyzed by RP-UPLC (reversed phase ultra performance liquid chromatography) either coupled to fluorescence (FLR) or electrospray ionization-ultra-high-resolution-quadrupole time-of-flight mass spectrometry (ESI-UHR-QTOF-MS)-based detection. For this purpose, two distinct instruments were used: an Acquity UPLC system (Waters Corporation, Milford, MA, USA) coupled to an Acquity Fluorescence Detector and to a maXis Impact ESI-QTOF MS (Bruker Daltonik GmbH, Bremen, Germany) (1), and a second Acquity UPLC system coupled exclusively to an Acquity Fluorescence Detector (2). The systems had a 740001685-TAP Acquity solvent manager (1 PM) and a sample manager (740001698-TAP (1 PM)). Sample injection was performed using a PLNO (Partial-Loop with Needle-Overfill) with an injection volume of 5 µL. The separation was achieved on a Waters Acquity UPLC HSS T3 (2.1 mm × 100 mm, 1.8 µm) column coupled to Acquity UPLC HSS T3 VanGuard (1.8 µm, 2.1 × 5 mm; Waters, Germany) pre-column, at 30 °C and a flow rate of 0.4 mL min^−1^. The initial mobile phase consisted of eluent A, 0.1% *v*/*v* formic acid in water; and B, 0.1% *v*/*v* formic acid in acetonitrile. The total analysis run time was 15 min/sample, where elution started with 99.9% A. From the third minute, this concentration dropped to 95% in favor of acetonitrile containing 0.1% formic acid (B). From the sixth to the tenth minute, the A ratio was 87.4 versus the 12.6 of solution B; consequently, the elution flow was maintained at 50–50 for 30 s. The last 4.5 min were for washing and reequilibration of the instrument for the following injection: the elution gradient was composed of 1% A and 99% B for two minutes and was later brought to 99.9% A and 0.1% B.

The detection was performed via a 740002848-TAP Acquity Fluorescence Detector (1 PM) set at an excitation wavelength of 280 nm, an emission wavelength of 320 nm with PMT gain of 1 and data rate of 1 (1) (2). The mass spectrometric detector was run in MS1 mode with an electrospray ionization interface in positive mode. The gas used for desolvation was nitrogen with the nebulizer pressure set at 8 bar. Dry gas flow was set at 8 L/min, whereas the temperature was maintained at 200 °C. Capillary voltage was set at 4000 V and target mass at 50–1000 *m*/*z* (2). LC-FLD-MS data from (1) was acquired via *Compass HyStar 3.2 SR2* Software and manually inspected using Compass DataAnalysis 4.4 SR1 package (Bruker Daltonik GmbH). Extracted ion chromatograms were made which corresponded to the [M+H]^+^ of each of the standards of interest (Appendix A). Peak area for each compound was curated manually and the values quantified using the QuantAnalysis 4.4 software (Bruker Daltonik GmbH, DE). In the case of gramine and AMI, the main fragment was not the protonated molecule but rather an in-source fragment with an m/z of 130.0666. Extracted ion chromatograms for this ion were also used to determine quantity. For (2), both data acquisition and instrument control were coordinated by *Empower 3* Software (Waters Corp., Eschborn, Germany).

### 4.6. Standards and Calibration Curves

A total of 10 mM stock solutions of gramine (GRA, 1.74 mg/mL), hordenine (HOR, 1.65 mg/mL), 3-Aminomethylindole (AMI, 1.46 mg/mL), tyramine (TYR, 1.37 mg/mL), noradrenaline (NOR, 1.69 mg/mL), dopamine hydrochloride (DOP, 1.89 mg/mL) and *N*-methyltryptamine (MeTRY, 1.74 mg/mL) were prepared in methanol:water (80:20). Solutions were further diluted in 0.1% formic acid to create 1 mM solutions. The 10 µM working standard solution was finally prepared by dilution as a mixture of NOR, DOP, HOR, TYR, AMI, GRA and Met-Try in 0.1% formic acid. This standard mix was stored in aliquots of 200 µL at −20 °C. Before use, it was brought to room temperature.

Calibration standards were constructed for both instruments by 3–5 replicated injections in the range 0.002–1500 µM for gramine, hordenine, AMI, tryptophan, tyramine (0.002, 0.005, 0.01, 0.02, 0.05, 0.1, 0.2, 0.5, 1, 2, 5, 10, 20, 50, 100, 500, 600, 700, 800, 900, 1000, 1250 and 1500 µM). These calibration curves were used to determine the limits of detection and quantification of both fluorescence and mass spectrometry detectors of the different instruments and to quantify analyte concentrations from plant tissues.

### 4.7. Method Validation

The method was validated following the IUPAC Harmonized Guidelines [86]. Performance tests on selectivity, accuracy and precision were conducted, along with measurements to establish the amplitude of the matrix effect and the carryover (2). Further injections helped us determine the sensitivity of the method and its limits of detection (1) (2). Moreover, we tested additional parameters such as reproducibility of the results and stability, i.e., the degradability of gramine over time.

Selectivity was determined as to how effective our method is in regard to discerning and measuring gramine. It was evaluated based on the clear separation of the gramine peak from the other components present in the standard mix and from the peaks of other metabolites detected in the barley and lupin tissues (2).

We evaluated accuracy as a measure of how close the measurements of a compound are to its true amount, whereas the precision of the method was assessed as how consistent are the results of repeated measurements performed on the same sample. Accuracy was measured in terms of percentage of gramine recovered by the analysis of barley lines which did not naturally contain gramine but were spiked with known amounts of the metabolite (*n* = 6). A total of 8, 25 and 100 pmol/µL was spiked in kernels extracts of Golden promise and ZDM 01467 lines for this purpose (2).

Precision was evaluated on the consistency of gramine concentrations retrieved in the replicates of the pooled seed samples of three lupin species producing gramine. Samples were run over a short period of time under similar conditions (2). Precision was not evaluated on the gramine measurements in pooled barley kernels as only one of the 22 accessions was producing it.

The abovementioned spiked samples were used also to evaluate the matrix effect: the results of these runs (*N* = 6) were compared to those of simple gramine standard injections at the same concentrations, paying attention to assessing possible traces of plant background material that might influence the peak expression and their detection. The peak area ratio was defined as the matrix effect.

Carryover of analytes in following chromatograms was evaluated by injecting ACN. Checks of the presence of previously analyzed compounds were performed (1) (2).

Method calibration was established by selecting the standards, generating calibration curves for both fluorescence detectors on instruments (1) and (2) and attributing a curve-fitting algorithm. With the injection of 3 replicates of different concentrations of the selected standards mix (containing gramine, hordenine, AMI, tyramine, noradrenaline, dopamine hydrochloride and *N*-methyltryptamine listed in Section 2.6), we generated calibration curves for gramine, AMI, tryptophan, hordenine and tyramine by displaying their peak area fluorescence response on a concentration against response plot. We determined their equations and assessed in which range they meet linearity requirements. The limit of detection (LOD) and the limit of quantification (LOQ) were assessed to have an indicator of the sensitivity of the method. LOD and LOQ were defined as the smallest concentration at which an analyte can be confidently determined as being present in a sample and the smallest concentration that can be reported as a quantitative value with acceptable precision and accuracy, respectively, and calculated [55]. LOD and LOQ were defined as follows:LOD = 3.3σ/S’,    LOQ = 10σ/S’,
are the calibration curves and S’ is the slope constituted by the peak area unit response per pmol/µL of concentration. Further evaluation of the method’s sensitivity was performed by directly comparing the limits of detection and quantification and linear ranges of UPLC-FLD (1) and UPLC-MS (1). 

Stability and reproducibility were evaluated as a measure of consistency of the results obtained from the analysis of the same samples on different occasions.

### 4.8. Figures

Plots were generated using R statistical Software (v. 4.0.0, R Development Core Team, 2020) and its complementary console R-studio (http://www.rstudio.com, accessed on 24 April 2023.).

## Figures and Tables

**Figure 1 plants-12-01930-f001:**
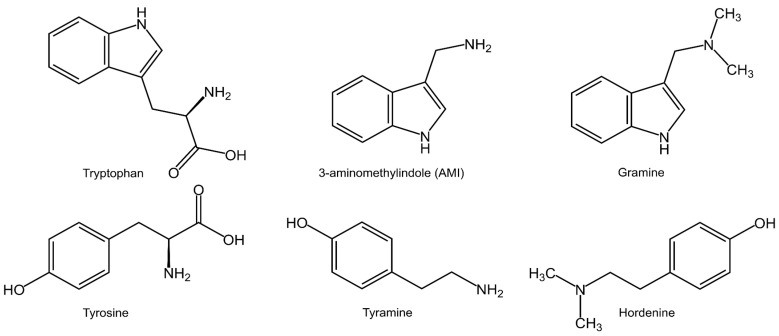
The chemical structures of tryptophan, 3-aminomethylindole (AMI), gramine, tyrosine, tyramine and hordenine.

**Figure 2 plants-12-01930-f002:**
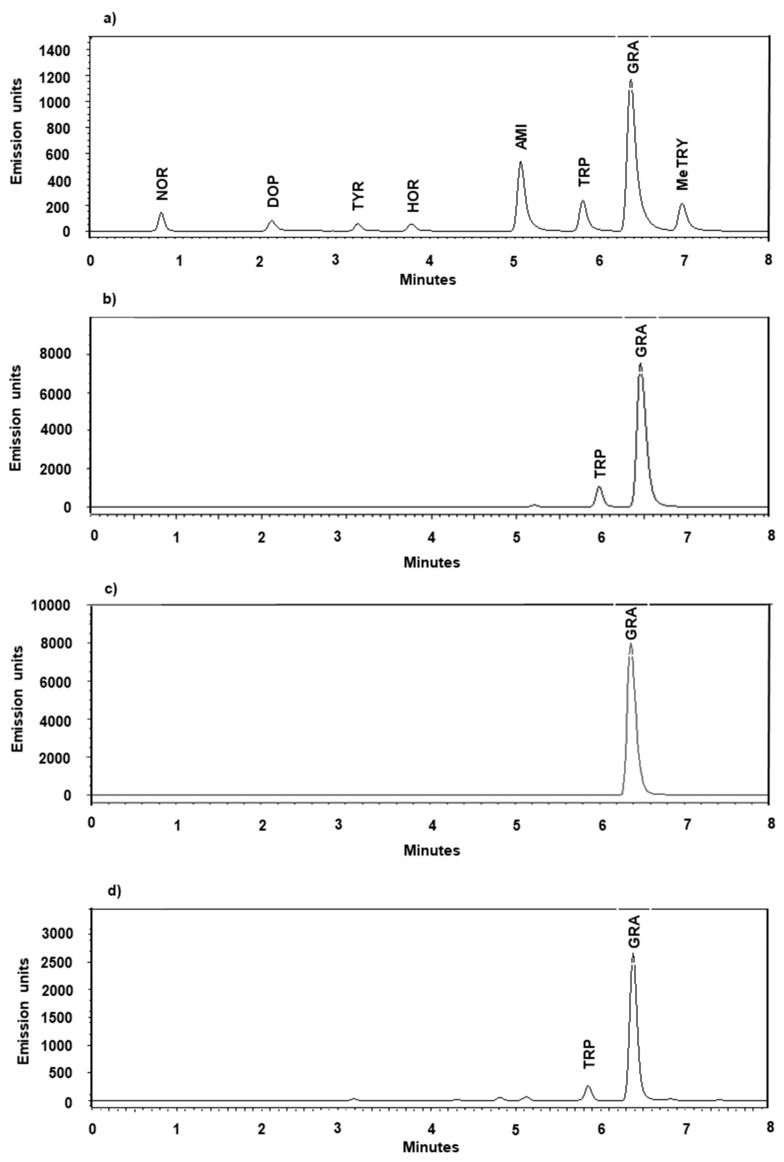
Representative UPLC-FLD chromatograms (2). (**a**) Standard mix analytes (noradrenaline (NOR), dopamine hydrochloride (DOP), tyramine (TYR), hordenine (HOR), 3-aminomethylindole (AMI), tryptophan (TRP), gramine (GRA) and *N*-methyltryptamine (MeTRY)), (**b**) barley sample (ONU 333 B3, leaf tissue, 8 DAG), (**c**) barley sample (ZDM 01467 kernel extraction) spiked with gramine (100 pmol/mg), (**d**) lupin sample (*L. mexicanus*, leaf tissue, 8 DAG).

**Figure 3 plants-12-01930-f003:**
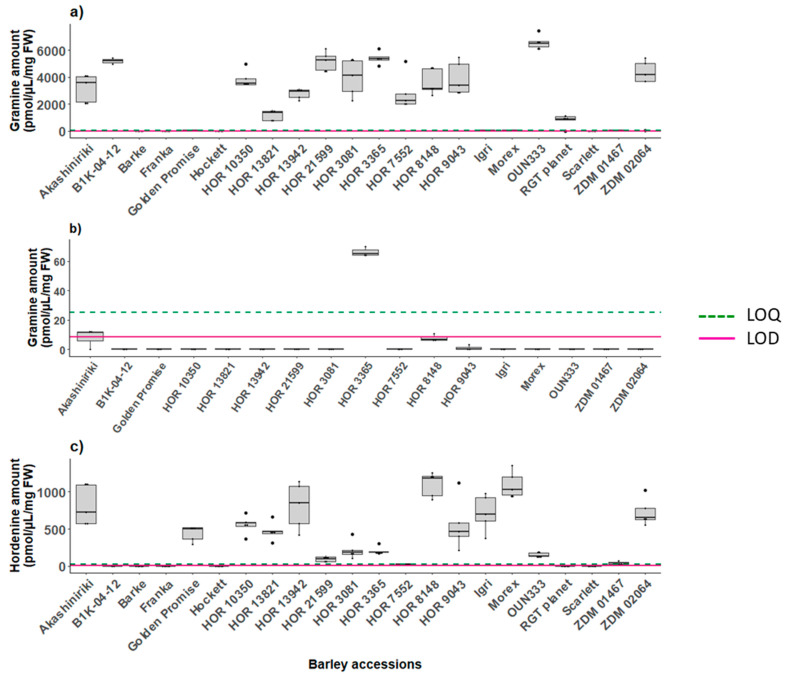
Analytes production in different barley tissues across 17–22 accessions: (**a**) gramine biosynthesis in leaf tissue (8 DAG) (*n* = 2–5), (**b**) gramine biosynthesis in kernels, (**c**) hordenine biosynthesis in root tissue (8 DAG) (*n* = 2–5). Graphs report the LOD and LOQ values reported in Appendix A.

**Figure 4 plants-12-01930-f004:**
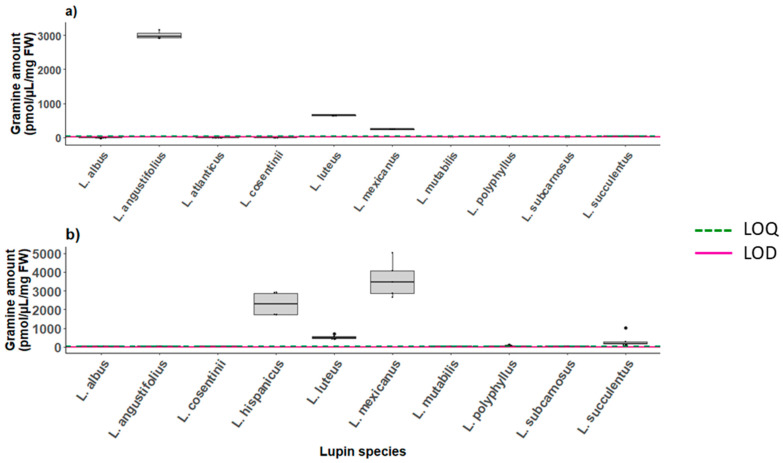
Analytes production in different lupin tissues across 9–10 species: (**a**) gramine biosynthesis in seeds (*n* = 6), (**b**) gramine biosynthesis in leaf tissue (*n* = 1–5). Graphs report the LOD and LOQ values reported in Appendix A.

**Table 1 plants-12-01930-t001:** Accuracy for gramine measurements (spiked samples and standard injections (*n* = 3–6)).

	Golden Promise			ZDM 01467			Gramine Standard		
Sample concentration (pmol/µL)	8	25	100	8	25	100	8	25	100
Replicate 1	8.500	29.06	112.56	9.170	29.31	123.5	8.660	26.19	99.86
Replicate 2	8.340	28.71	112.70	9.200	30.35	123.3	8.740	25.96	100.1
Replicate 3	8.740	28.55	112.72	9.220	30.82	123.5	8.720	26.10	99.34
Replicate 4	8.310	28.60	112.42	9.170	29.41	123.1	-	-	-
Replicate 5	8.430	28.62	112.59	9.230	30.46	123.5	-	-	-
Replicate 6	8.750	28.75	112.45	9.340	29.86	123.2	-	-	-
Mean	8.510	28.71	112.57	9.220	30.03	123.4	8.710	26.08	99.76
Standard deviation	0.19	0.18	0.12	0.06	0.61	0.18	0.04	0.12	0.38
Relative standard deviation	2.26	0.64	0.11	0.70	2.02	0.15	0.47	0.46	0.38

**Table 2 plants-12-01930-t002:** Precision of gramine quantifications (*n* = 3).

Injection	Response (Gramine Amount (pmol/µL/mg FW))		
	*L. luteus*	*L. angustifolius*	*L. mexicanus*
1	667	3000	251.5
2	671.2	3172	263.2
3	660.7	3065.11	232.2
Mean	666.3	3079	249.0
Standard deviation	5.31	86.5	15.7
RSD	0.80	2.81	6.28

**Table 3 plants-12-01930-t003:** Equations for the regression curves of the calibration injections on the UPLC-FLD (1) for gramine, hordenine, AMI, tryptophan and tyramine (*n* = 5).

Compound	Equation	SE y-Intercept	Coefficient of Determination R^2^
Gramine	y = 1,233,973.60x + 192,425.42	31,834.10	0.9965
Hordenine	y = 88,411.87x − 21,392.30	27,264.32	0.9963
AMI	y = 400,644.54x − 56,069.36	14,840.62	0.9973
Tryptophan	y = 148,118.67x − 77,669.45	6784.03	0.9981
Tyramine	y = 78,928.96x − 25,859.95	8311.62	0.9968

**Table 4 plants-12-01930-t004:** Equations for the regression curves of the calibration injections on the UPLC-FLD (1) for gramine, hordenine, AMI, tryptophan and tyramine (*n* = 5).

	Fluorescence Detection (1)		Mass Spectrometry (1)	
Compound	LOD (pmol/µL)	LOQ (pmol/µL)	LOD (pmol/µL)	LOQ (pmol/µL)
Gramine	0.09	0.26	0. 27 * 0.18 **	0.83 * 0.54 **
Hordenine	1.02	3.08	0.25	0.75
AMI	0.12	0.37	0.93	2.82
Tryptophan	0.15	0.46	0.33	1.00
Tyramine	0.35	1.05	0.48	1.46

* for EIC 130.0669; ** for EIC 175.1239.

**Table 5 plants-12-01930-t005:** Equations for the regression curves of the calibration injections on the UPLC-FLD (1) for gramine, hordenine, AMI, tryptophan and tyramine (*n* = 5).

Name	Common Name	Accession Number	IPK Genebank Code
*Lupinus subcarnosus* Hook	Texas bluebonnet	LUP 43	8839964
*Lupinus succulentus* Dougl. ex K. Koch	Succulent lupin or Arroyo lupin	LUP 48	8839965
*Lupinus cosentinii* Guss.	Hairy lupin	LUP 76	8839966
*Lupinus mutabilis* Sweet	(White) pearl lupin or Andean lupin	LUP 100	8839967
*Lupinus albus* L. *subsp. albus*	White lupin	LUP 211	8839368
*Lupinus luteus* L.	Yellow lupin	LUP 344	8839969
*Lupinus hispanicus* Boiss. & Reut.	Spanish lupin	LUP 549	8839970
*Lupinus polyphyllus* Lindl.	Garden lupin	LUP 562	8839971
*Lupinus mexicanus* Cerv. ex Lag.	Mexican lupin	LUP 574	8839972
*Lupinus atlanticus* Gladstones	-	LUP 5152	8839973
*Lupinus angustifolius* L. *subsp. angustifolius*	Narrowleaf lupin	LUP 5268	8839974

## Data Availability

The data presented in this study are available in the article or Appendix A. The raw MS files are available on request from the corresponding author.

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
