# Peer review of "A New Fluorescence Detection Method for Tryptophan- and Tyrosine-Derived Allelopathic Compounds in Barley and Lupin"

_plants, 2023, doi:10.3390/plants12101930_

Round 1
Reviewer 1 Report
1. Though some of the abbreviations are commonly used, it is still required to explain them once first met in the text, for example QTL, HPLC-MS/MS, GC-MS, GLC, LC-MS. Both HPLC-MS and LC-MS have the same meaning nowadays, better would to select one of these abbreviations and use it in a uniform way.
2. Page 3, last passage: “An alternative method to LC-MS that is reliable and accurate for quantification purposes is required for the analysis of gramine, hordenine and their related metabolites”
There is a lack of motivation in developing an alternative method to LC-MS for analysis of gramine, hordenine and their related metabolites. The authors have discussed a large number of methods used, but since the shortcomings were not revealed it doesn’t come evident why an alternative method should be developed. And why an alternative method should be exactly to LC-MS, why not GC-MS? A few explanations should be provided.
3. “Many solid phases were tested to achieve…”, “HSS-T3 solid phase with…” and elsewhere “solid” should be changed for “stationary”.
4. There is a reference to Fig 5 in chapter 2.7.2., but the figure itself is missing in the manuscript.
5. It is rather strange to use scanning mode with Q-TOF instrument when performing the quantitation procedure. This way the authors have intentionally decreased the sensitivity they could get with an LC-MS instrument of such type. It is well known that tandem mass spectrometry (MS/MS) especially when it’s high resolution mass spectrometry (HRMS) may significantly drop down LOD and LOQ for detection of particular compound or group of compounds. Hence, the discussed comparison of sensitivity achieved with Fluorescence Detector against MS is not appropriate.
6. Chapter 4.5
“In the case of gramine and AMI, the main fragment was not the molecular ion, rather a fragment with a m/z of 130.0666.”
Why do authors consider fragment ions anyway? Was in-source fragmentation or MS/MS applied? Some comments should be added to clarify this method procedure. Also, ESI usually doesn’t result in formation of molecular ions, while [M+H]+ is commonly called protonated molecule.
7. Check for spelling and syntax mistakes (lost spaces, dots) and use subscript format for formulae (for instance CHCl3)
Author Response
Dear Reviewers,
We are pleased to report that we have addressed all of your concerns and comments. We have significantly reduced the length of the introduction as well as made some comparisons of our study with previous research in the discussion section. The manuscript has been edited for English and syntax and we have added a glossary of abbreviations as requested.
Please see a detailed point-by-point response to the comments. Our responses are highlighted in blue text. Thank you very much.
Reviewer #1
- Though some of the abbreviations are commonly used, it is still required to explain them once first met in the text, for example QTL, HPLC-MS/MS, GC-MS, GLC, LC-MS. Both HPLC-MS and LC-MS have the same meaning nowadays, better would to select one of these abbreviations and use it in a uniform way.
Thank you for this comment. In an effort to make things uniform and more clear, we disclosed abbreviations and added a glossary section (line 31).
- Page 3, last passage: “An alternative method to LC-MS that is reliable and accurate for quantification purposes is required for the analysis of gramine, hordenine and their related metabolites”
There is a lack of motivation in developing an alternative method to LC-MS for analysis of gramine, hordenine and their related metabolites. The authors have discussed a large number of methods used, but since the shortcomings were not revealed it doesn’t come evident why an alternative method should be developed. And why an alternative method should be exactly to LC-MS, why not GC-MS? A few explanations should be provided.
We have listened to the concerns of the reviewer and have made the text regarding our motivation behind developing this method as clear as possible and have made clarifications in the text. Please see the applied corrections at lines 92-98.
- “Many solid phases were tested to achieve…”, “HSS-T3 solid phase with…” and elsewhere “solid” should be changed for “stationary”.
Thank you for the comment. The terms have been substituted according to the reviewer’s wishes (see lines 114, 137, 251).
- There is a reference to Fig 5 in chapter 2.7.2., but the figure itself is missing in the manuscript.
The reviewer is absolutely correct. The text has been corrected to reflect our reference to figure 4 (line 236 & 239). Thank you for pointing that out.
- It is rather strange to use scanning mode with Q-TOF instrument when performing the quantitation procedure. This way the authors have intentionally decreased the sensitivity they could get with an LC-MS instrument of such type. It is well known that tandem mass spectrometry (MS/MS) especially when it’s high resolution mass spectrometry (HRMS) may significantly drop down LOD and LOQ for detection of particular compound or group of compounds. Hence, the discussed comparison of sensitivity achieved with Fluorescence Detector against MS is not appropriate.
We thank the reviewer for their comment. MS/MS experiments are generally the way one would want to quantify compounds on an LC-MS system, especially when the mass spectrometer used has a large dynamic linear range, e.g. a triple quadropole based system or a HRMS system like an orbitrap in which ions can be selected and highly enriched. We did not test LOD or LOQ on either of these types of systems, but rather a Bruker Maxis qTOF instrument with an orthogonal TOF. As per the manufacturer’s training material (Revision D, January 2019) and the user’s handbook for the instrument regarding:
“Quantitation - MS/MS versus MS - In (orthogonal) TOF mass spectrometers, quantitation is typically carried out in full scan MS mode. As the mass accuracy is very high you can easily distinguish between compounds of similar mass (mass window 5 mDa) and these EIC traces contain only very low noise. In MSMS mode you would lose signal intensity during fragmentation. Only in very complex samples MSMS can be more sensitive because of noise reduction.”
We nonetheless performed MS/MS multiple reaction monitoring (MRM) experiments with the compounds in this study following the reviewer’s comment and found that on a Maxis I or Maxis II orthogonal TOF instrument, quantification by full scan MS mode gave at least a full order of magnitude higher sensitivity in our typical sample matrices than using an MRM based protocol. Given the in-source fragmentation already apparent for these compounds in combination with the higher sensitivity we obtained with two qTOF instruments in MS scanning mode, we have chosen to keep the comparison in the manuscript. Out of respect for the reviewer’s comment, we have qualified our comparison to reflect those concerns.
- Chapter 4.5
“In the case of gramine and AMI, the main fragment was not the molecular ion, rather a fragment with a m/z of 130.0666.”
Why do authors consider fragment ions anyway? Was in-source fragmentation or MS/MS applied? Some comments should be added to clarify this method procedure. Also, ESI usually doesn’t result in formation of molecular ions, while [M+H]+ is commonly called protonated molecule.
We are sorry for any confusion caused in the text. We consider fragment ions because AMI undergoes in-source fragmentation following ionization. The resulting major ion observed is a fragment with a m/z of 130.0666. We have corrected the text to make clear we are speaking of an in-source fragment (see line 478).
- Check for spelling and syntax mistakes (lost spaces, dots) and use subscript format for formulae (for instance CHCl3).
We thank the reviewer and have double checked the syntax. Thank you for the hint.
Reviewer 2 Report
The manuscript looks very thorough; it presents the results of rather detailed work, namely new technique of HPLC with fluorescence detection method for tryptophan- and tyrosine-related compounds (indolic and phenolic amines) in barley and lupin. The reported results are well validated (as it is discussed at page 10); some of them demonstrated the excellent accuracy. For instance, Table 1 includes the results of gramine measurements; the relative standard deviations of most of them are less than 1%.
Hence, it is not surprising that critical comments appeared to be highly insignificant.
- Pages 3, 11, and 12 contain the text only and no Tables or figures; hence they are slightly difficult for reading. However, the reviewer has no idea how to improve this situation.
- The chromatographic peak of gramine (Fig. 2c) looks non-symmetrical (its tail is distorted). So far as it is not the overlapping with another component, maybe a note worth to be added that it is the results of overloading the column (or not?).
- There are no problems with presenting the results in Table 1. However, they are in Table 2. At first, not necessary to indicate the standard deviations with 3-4 significant digits, like 5.31, 86.52, and 15.65. In these cases only one-two digits should be presented. Finally all values should be rounded up as 666 +- 5, 3080 +- 90, and 249 +- 16.
- The values 2308.65, 3604.81, and 515.41 at the page 9 (in the text) contain 5-6 significant digits. It means that the Authors have used the extra précised analytical technique which provides the relative precision of results at the level of 10-3 – 10-4%. So far as it is impossible, these results should be rounded up to 2310, 3600, and 515 pmol/mkL
After correction of moments mentioned the manuscript can be recommended for publication.
Author Response
Dear Reviewers,
We are pleased to report that we have addressed all of your concerns and comments. We have significantly reduced the length of the introduction as well as made some comparisons of our study with previous research in the discussion section. The manuscript has been edited for English and syntax and we have added a glossary of abbreviations as requested.
Please see a detailed point-by-point response to the comments. Our responses are highlighted in blue text. Thank you very much.
Reviewer #2
The manuscript looks very thorough; it presents the results of rather detailed work, namely new technique of HPLC with fluorescence detection method for tryptophan- and tyrosine-related compounds (indolic and phenolic amines) in barley and lupin. The reported results are well validated (as it is discussed at page 10); some of them demonstrated the excellent accuracy. For instance, Table 1 includes the results of gramine measurements; the relative standard deviations of most of them are less than 1%.
Hence, it is not surprising that critical comments appeared to be highly insignificant.
- Pages 3, 11, and 12 contain the text only and no Tables or figures; hence they are slightly difficult for reading. However, the reviewer has no idea how to improve this situation.
We apologize for the formatting. Hopefully the final layout will be more readable.
- The chromatographic peak of gramine (Fig. 2c) looks non-symmetrical (its tail is distorted). So far as it is not the overlapping with another component, maybe a note worth to be added that it is the results of overloading the column (or not?).
Replacement of the pre-column solved the distortion of peaks. A new chromatogram resulting from new runs of spiked samples has been added to the manuscript (see line 149). Thank you.
- There are no problems with presenting the results in Table 1. However, they are in Table 2. At first, not necessary to indicate the standard deviations with 3-4 significant digits, like 5.31, 86.52, and 15.65. In these cases only one-two digits should be presented. Finally all values should be rounded up as 666 +- 5, 3080 +- 90, and 249 +- 16.
We thank the reviewer for pointing out the importance of significant figures. We have corrected all numbers in both text and main tables: we corrected all amounts to have 4 sig figs and all standard deviations to have 3 sig figs.
- The values 2308.65, 3604.81, and 515.41 at the page 9 (in the text) contain 5-6 significant digits. It means that the Authors have used the extra précised analytical technique which provides the relative precision of results at the level of 10-3 – 10-4%. So far as it is impossible, these results should be rounded up to 2310, 3600, and 515 pmol/mkL
We thank the reviewer for pointing out the importance of significant figures. We have corrected all numbers in both text and main tables: we corrected all amounts to have 4 sig figs and all standard deviations to have 3 sig figs.
After correction of moments mentioned the manuscript can be recommended for publication.
Reviewer 3 Report
The main object of this research are barley and lupins, crops of economic importance. The researchers are focused on the development and validation of a new method for quantification of two plant byproducts – the alkaloids gramine and hordenine. They applied a modern analytical method and technique like UPLC with fluorescence detection.
The paper is not well written, the text is not clear and easy to read. It's as if the authors pulled out pieces of a thesis and roughly stitched the article together. However, the authors have done a great deal of work in the validation step, but the manuscript must be rewritten.
I have some questions and suggestions:
General:
· The authors made a single-laboratory validation (SLV). It establishes and documents the performance characteristics of a method, thereby demonstrating whether the method is suitable for a particular analytical purpose. So, the validation of a quantity method usually follows the instructions of an accepted protocol, e.g. ICH- guidelines of the Federal Register or IUPAC Harmonized Guidelines or equal. Which guidelines did the researchers follow in the current validation study? Please, cite the source of the validation guidelines protocol followed.
· What is new on this method? Is it the detection step or the chromatographic step? It isn’t clear. In the Discussion the authors said “Accurate chromatographic separation was achieved after testing various combinations of different solid phases and eluents”, but in the whole manuscript isn’t any data (in the Result) about the choice of the chromatographic conditions made. There is only refer to Figure S2 in the 2.1. Method development.
· There isn’t any data about the origin of the extraction step. Is it a new extraction procedure, or not? Reference is required.
· What is about statistical analysis? Is any statistic method applied? In the quantification of any parameters measured in biological materials, the data is processed statistically.
· There are many abbreviation in the main text, so for better text understanding und reading, I suggest the author to add an abbreviation interpretation section.
1. Intoduction
· In my opinion, the Introduction is too long (3 pages!). An optimal length would be 1-1.5 pages. For example, the info about the distribution of these alkaloids in the plant world or the path of the biosynthesis of this alkaloids are unnecessary. I think, it would suffice to say that these alkaloids have a protective effect and are also present in barley and lupins, and perhaps in what quantities. Тhe authors can only give the primary and intermediate metabolites of their synthesis without going into details.
· The info about the methods for qualification and quantification of this alkaloids are also too long, the manuscript isn’t review. The authors must be give only an info to introduce the necessary of a new better method for quantification of this byproducts.
· A lot of info can be moved into the discussion section, e.g. why the HPLC technique was chosen.
2. Results. Figure 3.
· What is mean with “fist extraction” in the foot note?
2.1. Method development: “This project was originally started to study the expression of gramine and hordenine in barley….”
· It is not an oral presentation, here must be written exactly what is done: how are chosen the conditions and technique of the extraction step, eluents, HPLC conditions, etc. The choice made must be justified and supported by concrete evidence: what parameters were tested, what were the results.
2.3. Selectivity, accuracy, matrix effect and precision
“Visual observation accompanied by comparison of retention times and peak shape…”
· This isn’t the correct approach, and is not correct to draw any conclusion about the “efficient chromatographic separation”.
4. Discussion:
· In the discussion section, I suggest the authors also compare the other validation parameters with their of the conventional, and other methods before concluding how reliable this new method is, like accuracy, selectivity, matrix effect, recovery etc.
Author Response
Dear Reviewers,
We are pleased to report that we have addressed all of your concerns and comments. We have significantly reduced the length of the introduction as well as made some comparisons of our study with previous research in the discussion section. The manuscript has been edited for English and syntax and we have added a glossary of abbreviations as requested.
Please see a detailed point-by-point response to the comments. Our responses are highlighted in blue text. Thank you very much.
General:
- The authors made a single-laboratory validation (SLV). It establishes and documents the performance characteristics of a method, thereby demonstrating whether the method is suitable for a particular analytical purpose. So, the validation of a quantity method usually follows the instructions of an accepted protocol, e.g. ICH- guidelines of the Federal Register or IUPAC Harmonized Guidelines or equal. Which guidelines did the researchers follow in the current validation study? Please, cite the source of the validation guidelines protocol followed.
We thank the reviewer for this very important observation and question. We followed the IUPAC Harmonized Guidelines along with the book “introduction to modern liquid chromatography” by Snyder, Kirkland and Dolan. The manuscript has been corrected. See line 495.
- What is new on this method? Is it the detection step or the chromatographic step? It isn’t clear. In the Discussion the authors said “Accurate chromatographic separation was achieved after testing various combinations of different solid phases and eluents”, but in the whole manuscript isn’t any data (in the Result) about the choice of the chromatographic conditions made. There is only refer to Figure S2 in the 2.1. Method development.
Thank you very much for bringing up this point. We propose both new detection and chromatographic steps (see lines 97-98). We added further details (including figures) to our method development section in order to address your concerns, Please see lines 114 - 141.
- There isn’t any data about the origin of the extraction step. Is it a new extraction procedure, or not? Reference is required.
We apologize for the omission. The information has been added (see line 417).
- What is about statistical analysis? Is any statistic method applied? In the quantification of any parameters measured in biological materials, the data is processed statistically.
No statistical tests were applied to the data generated. However, replicated data was subjected to analysis by calculating standard deviation and relative standard deviation as is appropriate.
- There are many abbreviation in the main text, so for better text understanding und reading, I suggest the author to add an abbreviation interpretation section.
Thank you for this comment. In an effort to make things uniform and more clear, we disclosed abbreviations and added a glossary section (line 31).
- Intoduction
- In my opinion, the Introduction is too long (3 pages!). An optimal length would be 1-1.5 pages. For example, the info about the distribution of these alkaloids in the plant world or the path of the biosynthesis of this alkaloids are unnecessary. I think, it would suffice to say that these alkaloids have a protective effect and are also present in barley and lupins, and perhaps in what quantities. Тhe authors can only give the primary and intermediate metabolites of their synthesis without going into details.
In order to reduce and consolidate the introduction, we have edited the sections pertinent to the biosynthesis of gramine.
The biosynthesis section has been partially removed, and information has been moved to the discussion as suggested ( see lines 64-68 and lines 288-331).
- The info about the methods for qualification and quantification of this alkaloids are also too long, the manuscript isn’t review. The authors must be give only an info to introduce the necessary of a new better method for quantification of this byproducts.
We have reduced the introduction to reflect the reviewer’s wishes. This section contains less background regarding previous studies.
- A lot of info can be moved into the discussion section, e.g. why the HPLC technique was chosen.
We appreciate the comments regarding the flow of the manuscript. We have therefore shortened this section considerably and moved relevant themes to the discussion where appropriate.
- Results. Figure 3.
- What is mean with “fist extraction” in the foot note?
We are sorry for any confusion caused in the text. Both the text and the legends are now referring to the various sample collections as “8 days after germination (8 DAG)” and “11 days after germination (11 DAG)”.
2.1. Method development: “This project was originally started to study the expression of gramine and hordenine in barley….”
- It is not an oral presentation, here must be written exactly what is done: how are chosen the conditions and technique of the extraction step, eluents, HPLC conditions, etc. The choice made must be justified and supported by concrete evidence: what parameters were tested, what were the results.
The sentence has been rephrased (line 110) and more details about the method development have been added (please see lines 114 - 141).
2.3. Selectivity, accuracy, matrix effect and precision
“Visual observation accompanied by comparison of retention times and peak shape…”
- This isn’t the correct approach, and is not correct to draw any conclusion about the “efficient chromatographic separation”.
We have decided to remove this statement in response to the reviewer’s comments.
Discussion:
- In the discussion section, I suggest the authors also compare the other validation parameters with their of the conventional, and other methods before concluding how reliable this new method is, like accuracy, selectivity, matrix effect, recovery etc
In combination with reducing the introduction according to the comments of the reviewers, we have complied with the suggestion and the discussion now includes the requisite comparisons. Thank you for the suggestion.
Round 2
Reviewer 1 Report
All needed corrections were done
Reviewer 3 Report
Dear Authors,
you have done great work.
I wish you success!